# HARD2VERIFY: A STEP-LEVEL VERIFICATION BENCHMARK FOR OPEN-ENDED FRONTIER MATH

## ABSTRACT

Large language model (LLM)-based reasoning systems have recently achieved gold medal-level performance in the IMO 2025 competition, writing mathematical proofs where, to receive full credit, each step must be not only correct but also sufficiently supported. To train LLM-based reasoners in such challenging, open-ended settings, strong verifiers capable of catching step-level mistakes are necessary prerequisites. We introduce *Hard2Verify*[1], a human-annotated, step-level verification benchmark produced with over 500 hours of human labor. Hard2Verify is designed to rigorously assess step-level verifiers at the frontier: Verifiers must provide step-level annotations or identify the first error in responses generated by frontier LLMs for very recent, challenging, and open-ended math questions. We evaluate 29 generative critics and process reward models, demonstrating that, beyond a few standouts, open-source verifiers lag closed source models. We subsequently analyze what drives poor performance in step-level verification, the impacts of scaling verifier compute, as well as fundamental questions such as self-verification and verification-generation dynamics.

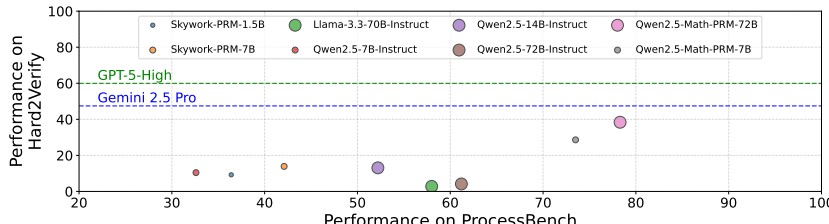

Figure 1: Comparison of models evaluated on both ProcessBench and Hard2Verify. Past benchmarks do not sufficiently evaluate in the frontier-level math settings that Hard2Verify does; On the same error identification task, Qwen2.5-Math-PRM-72B performance drops from ProcessBench state-of-the-art at 78.3 to 38.4 on Hard2Verify.

## 1 INTRODUCTION

Mathematical reasoning serves as a gold-standard evaluation setting for benchmarking reasoning progress in large language models (LLMs). Over the past half-decade, benchmarks have been introduced to assess LLMs at the grade-school (Cobbe et al., 2021), high-school (Hendrycks et al., 2021), university (Zhang et al., 2023), and competition math level (MMA, 2025; He et al., 2024a; Gao et al., 2024). However, the progress of mathematical reasoning ability of LLMs has outpaced benchmark creation, with every subsequent release of a frontier LLM saturating new benchmarks, most recently with GPT-5 Pro achieving 96.5%+ on AIME 2024. As a result, recent efforts (Glazer et al., 2024; Phan et al., 2025) have written novel, unseen mathematical questions to test LLMs.

While the training approaches of closed frontier models remain a secret, open-source progress in mathematical reasoning has been driven by scaling *reinforcement learning from verifiable rewards* (RLVR) (Lambert et al., 2024), with the breakthrough of DeepSeek-R1 (Guo et al., 2025) leading to an explosion of interest. This paradigm requires training data with solutions that are easily *verifiable*,

---

[1]We will open-source the benchmark, including evaluation code.

Table 1: Comparison between Hard2Verify and existing step-level math benchmarks.

| | Question Difficulty | Open-Ended Responses? | Natural Responses? | Generator Strength | Annotator | Step-Level Labels? |
|---|---|---|---|---|---|---|
| MR-GSM8K (Zeng et al., 2023) | Easy | ✗ | ✓ | Weak | Human | ✓ |
| MR-MATH (Xia et al., 2025) | Easy | ✗ | ✓ | Weak | Human | ✓ |
| MR-Ben (Zeng et al., 2023) | Easy | ✗ | ✓ | Weak | Human | ✓ |
| ProcessBench (Zheng et al., 2024) | Easy-Hard | 10.3% | ✓ | Weak-Medium | Human | ✗ |
| PRMBench (Song et al., 2025) | Easy | ✗ | ✗ | Weak | Synth. + Human Check | ✓ |
| Hard2Verify (Ours) | Hard | 78.5% | ✓ | Strong | Human | ✓ |

i.e., have solutions that can be easily checked against a known ground-truth by string matching or symbolic checkers. Math benchmarks, for the most part, also adopt the verifiable setup, where a model response is considered correct if its final answer matches the established ground-truth. Answer correctness, while a necessary condition for overall solution correctness, is not sufficient: It is now established that LLMs can produce incorrect intermediate reasoning but conclude with correct final answers (Lightman et al., 2023; Zheng et al., 2024; Setlur et al., 2025).

The next frontier for LLMs is solving problems that are *hard to verify*. A grand example such a problem is proving the Riemann hypothesis, where the expected solution is not a short phrase, but a multi-step proof. To verify correctness, each step must be rigorously checked. Hints of open-ended problem solving abilities already exist: advanced reasoning systems (OpenAI, 2025; Google, 2025b; Huang & Yang, 2025) have achieved gold-level performance in the 2025 IMO. Here, LLM outputs were judged at the step-level by human experts who determined if steps are both correct and sufficiently supported, with supporting lemmas and claims all appropriately stated and applied.

Training reasoning models capable of open-ended problem solving requires scalable *automatic evaluation*: Not every LLM rollout during RLVR training can be audited by human experts. Rather, evaluation in open-ended settings requires *step-level verifiers*, typically process reward models (PRMs) or generative critic models. Such verifiers have already been used to train policy models with dense process rewards (Lightman et al., 2023; Shao et al., 2024; Zha et al., 2025). Furthermore, step-level verifiers are also used in many test-time scaling methods, selecting the most promising candidate from multiple solutions or steps (Snell et al., 2024; Yu et al., 2025; Lifshitz et al., 2025). However, are these step-level verifiers sufficient for pushing the frontier of mathematical reasoning?

This work introduces *Hard2Verify*, which gauges the ability of step-level verifiers to push the frontier. *Hard2Verify* benchmarks verifiers in assessing *frontier* LLM responses to difficult, recent, and open-ended math problems. We curate challenging problems from recent international mathematics competitions like IMO and Putnam used to sample responses from three top-tier LLMs, GPT-5 (high) (OpenAI, 2025a), Gemini 2.5 Pro (Google, 2025a), and Claude Sonnet 4 (thinking) (Anthropic, 2025), representing various frontier points in reasoning LLMs. Finally, we employ PhD-level math experts to annotate each model-generated step. The resulting benchmark is the culmination of over 500 hours of human effort, passing three rounds of independent agreement checks. This meticulous process yields 1860 rigorously graded responses across 200 unique model responses.

Beyond operating at the frontier, *Hard2Verify* distinguishes itself from existing benchmarks for step-level annotation. First, we emphasize collecting open-ended questions, with 78.5% of our samples being open-ended. This way, verifiers cannot "cheat" if they have seen the question or ground-truth answer during training; rather verifiers must substantively assess each step to determine correctness. Second, step correctness is judged not only on correctness, but also based on whether all invoked results, such as supporting lemmas or claims, are correctly stated and applied; saying "$X$ follows from $Y$" receives no credit if $Y$ is not sufficiently justified or properly invoked. Third, *Hard2Verify* focuses on benchmarking verifiers in naturally occurring application settings: Verifiers must assess *model-written* responses, which often differ dramatically from human-written reference answers.

We benchmark 29 models spanning proprietary models to open-weight models to PRMs. Compared to past work, *Hard2Verify* represents a step up in difficulty, as shown in Fig. 1; Models capable of scoring 60%+ on ProcessBench (Zheng et al., 2024) are unable to crack 20% on *Hard2Verify*. Our analysis reveals that this degraded performance is because weaker verifiers cannot identify mistakes, marking nearly *every* step as correct. We additionally analyze several fundamental questions in step-level verification: How should one to scale verifier compute? What are the impacts of self-verification? How much easier is generation than verification for frontier models?

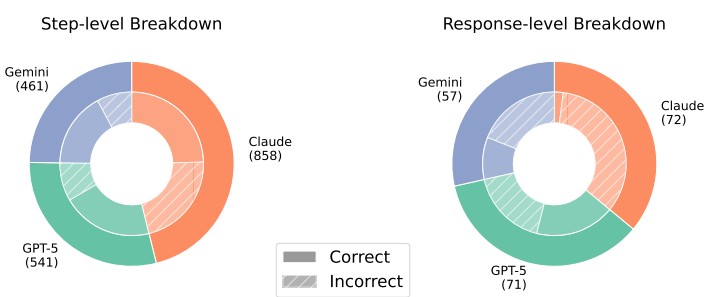

Figure 2: Breakdown of correct vs. incorrect steps (left) and responses (right) by model. We consider a response incorrect if *any* step in the response is labeled incorrect.

## 2 BACKGROUND AND RELATED WORK

**LLM-based verification.** To meet demands for scalable evaluation, LLM-based evaluators have been proposed, originally focusing on chat settings (Zheng et al., 2023). However, as LLMs are deployed in challenging reasoning settings (Ke et al., 2025), the need for more capable evaluators has grown. To get denser evaluation signal, focus quickly shifted to PRMs (Lightman et al., 2023) and synthetic ways to curate step-level training data (Wang et al., 2023; Luo et al., 2024). However, when used as dense reward signals for policy optimization, recent work has shown only limited improvement over outcome-level counterparts (Shao et al., 2024), which results from shortcomings process reward formulations. PRMs only measure if a step *could* lead to a correct, likely short-form final answer, not whether the step is correct in any absolute sense. As a result, recent focused has shifted towards *generative verifiers* (Mahan et al., 2024; Zhang et al., 2025a), using the natural language generation abilities of LLMs to perform verification. This allows for more precise description of evaluation criteria while enjoying benefits of increased inference-time compute.

**Benchmarking step-level verifiers in math settings.** Table 1 contrasts *Hard2Verify* with related benchmarks. MR-GSM8K (Zeng et al., 2023) annotate model responses to GSM8K (Cobbe et al., 2021) questions on a per-step basis to evaluate generative models as evaluators. MR-MATH (Xia et al., 2025) and MR-Ben (Zeng et al., 2024) follow similar approachs, increasing question difficulty with slightly harder sources like MATH (Hendrycks et al., 2021) and MMLU (Hendrycks et al., 2020). The two most relevant works to *Hard2Verify* are ProcessBench (Zheng et al., 2024) and PRMBench (Song et al., 2025). ProcessBench uses a mix of easy (GSM8K and MATH) and hard (OlympiadBench and Omni-MATH) questions, but is comprised largely of samples with single answer outputs. Further, ProcessBench only evaluates first error identification ability of verifiers, rather than tasking verifiers to evalute *every* step. PRMBench, on the other hand, obtains step-level annotations by taking *human-written* and model-generated solutions from PRM800K and injecting errors with an LLM, yielding responses that are *not naturally occurring*: Human-and model-written text may have large differences in style and substance, while injected errors may not represent naturally occurring errors in model generation. Hard2Verify, in contrast, operates at the current frontier, tasking verifiers to evaluate responses from frontier-level LLMs to difficult open-ended questions.

## 3 THE HARD2VERIFY BENCHMARK

### 3.1 DESIGN PHILOSOPHY

Hard2Verify is designed to test verifiers at the frontier of LLM-based math reasoning. At the question, response, and annotation level, Hard2Verify is curated based on the following philosophy:

- **Questions.** To measure progress in step-level verification, we must characterize how verifiers perform on *extremely difficult, open-ended* math questions. Open-ended problems represent the next frontier of mathematical reasoning, one where verifiers become increasingly important in lieu of available ground-truth answers. We focus our data collection on very recent mathematical Olympiads, prioritizing open-ended questions.
- **Model responses.** The responses that verifiers evaluate must be from *highly capable, frontier-level models.* To push the frontier of math reasoning, verifiers must be able to tell when the most

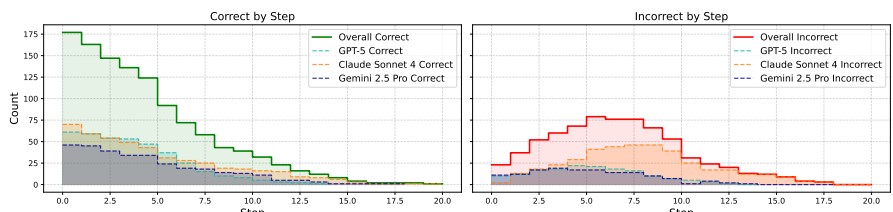

Figure 3: Count of correct and incorrect labels by model solution step.

powerful models make potentially subtle mistakes. Moreover, such mistakes should be *naturally occurring*, i.e., arise naturally from the model generation process. We do not inject or edit an existing correct model-or human-written solution. This is meant to closely approximate the response distribution that verifiers will see "in the wild", as they are applied in frontier math settings.

- **Annotation process.** We employ a *strict view* of response grading: Any step that contains a mistakes or is derived from a previous mistake is considered incorrect, i.e., we do not employ "Error Carried Forward" grading. This is inspired by competitive math settings, the entire solution must be correct to receive full points.

Based on this philosophy, we create Hard2Verify, as we describe in detail next.

### 3.2 CURATING HARD QUESTIONS

We construct our benchmark by collecting problem statements and official solutions $(Q, A_{\text{official}})$ from leading math competitions including the IMO, Putnam, and INMO; We provide a full list of sources in App. B. We focus question curation on recent (2024 and beyond) Olympiad-level math competitions. For each Olympiad, we parse the official PDFs using MathPix and extract all content in LaTeX to preserve mathematical typography and ensure stable equation rendering. We exclude image-dependent problems and only keep questions that could be solved using textual information. The resulting question set comprises 80 frontier-level problems from 10 distinct Olympiads.

### 3.3 RESPONSE GENERATION

Using our curated question pool, we sample responses from three frontier LLMs: GPT-5 (with high reasoning), Gemini 2.5 Pro, and Claude Sonnet 4 (Thinking). We employ a standardized prompt (App. D), instructing models to produce exam-style, stepwise proofs that mirror how an Olympiad participant would structure a solution. We use the same prompt template and decoding settings across models and disable access to external tools, like web search or code interpreters. Each model produces a single solution per problem, which we record for downstream evaluation. These samples are challenging; for example, Gemini 2.5 Pro takes up to 15 minutes to return a solution via API access. After curating all model responses to all questions, we filter out responses with undesirable qualities, such as a small number of long, dense steps or responses with degenerate outputs. This leaves us with a compact but high quality set of 200 responses.

### 3.4 ENSURING HIGH-QUALITY ANNOTATIONS

After sampling responses to our curated questions, human annotators meticulously annotate each model solution step-by-step. We partnered with [Data Co.] (Redacted for Review), a data annotation company. [Data Co.] employs mathematical experts, with a super-majority of our annotators having an advanced graduate level education in mathematics. To ensure consistent and high quality evaluations, we provided comprehensive annotation instructions as well official solutions $A_{\text{official}}$ as references. Annotation began with a multi-round pilot study, where we hand-annotated three model responses, then worked together with annotators to review samples, solicited feedback from annotators, and finetuned evaluation instructions accordingly. We then performed annotations in batches of samples, performing spot-checks of samples as they became available. This is in addition to internal processes at [Data Co.], which include initial human annotation and three rounds of human review, where annotations were reviewed for correctness and guideline alignment. Overall, *this process represents over 500 hours of manual human labor*. See App. E for more annotation details.

### 3.5 OVERALL DATASET STATISTICS.

Our annotation process yields 1,860 unique model steps annotated across 200 model solutions. 62% (1,154/1,860) steps are labeled correct, while the remaining 706 are labeled incorrect. Fig. 2 shows how models perform on a step-level and problem level. We consider a model response correct if *all* steps in the solution are graded correct by humans. Claude Sonnet 4 takes the most steps but gets the least percentage of steps correct, whereas GPT-5 and Gemini 2.5 Pro are the best performing model in terms of step-level accuracy. However, at the response level, GPT-5 outperforms Gemini 2.5 Pro by larger margins. Claude Sonnet 4, while achieving over 50% step-level accuracy, fails to string correct steps together, only producing 4 entirely correct solutions out of 72. Fig. 3 visualizes how errors appear as a function of steps, with all three models following similar trends: Errors tend to occur in the middle of model solutions appearing after a few solution steps.

### 3.6 EVALUATION TASKS

Our step-level annotations enable us to benchmark verifiers on three distinct tasks: (1) Step-level correctness (`Step-Level`), (2) Response-level correctness (`Response-Level`), and (3) First error identification (`ErrorID`). The `Step-Level` task corresponds to the setup in (Song et al., 2025), whereas the `ErrorID` tasks corresponds to that of Zheng et al. (2024). As we show in § 4, both tasks are challenging settings for current verifiers. We provide our evaluation prompts in App. D.

**`Step-Level`**. Here, the verifier is tasked with determining the correctness of each step. Generative verifiers are prompted to output a binary yes/no label for each step, whereas PRM step-level scores are converted to binary labels via a fixed threshold.

**`Response-Level`**. We also consider a response outcome task derived from `Step-Level` labels and predictions. This task reflects a strict grading of open-ended math problems: For a question to be correct, *all* steps in the solution must be deemed correct. Therefore, if *any* step in the solution is incorrect, we consider the solution wrong. From human labels, we create a single overall response-level correctness label. Likewise, given step-level predictions from a verifier, we create a response-level prediction. Note that this setting is less strict than the `Step-Level` setup: Exact step labels need not match exactly for a verifier to agree with a human at the response level.

**`ErrorID`**. Here, the verifier is prompted to output the first step which contains a mistake in the model solution, if present. If no error is present, a generative verifier may output step -1, corresponding to "No error". For PRMs, we select the first step below the correctness threshold.

## 4 EXPERIMENTS

### 4.1 EVALUATION METRICS

Let TPR and TNR denote the True Positive Rate and True Negative Rate, i.e., verifier accuracies on the correct and incorrect samples, respectively. We define *Balanced Accuracy* as the mean of TPR and TNR and *Balanced F1 Score* as the harmonic mean of TPR and TNR[2]:

$$\text{Balanced F1 Score} = \frac{2\,\text{TPR} \cdot \text{TNR}}{\text{TPR} + \text{TNR}}, \tag{1}$$

We report Balanced Accuracy and Balanced F1 Score for all tasks. The ground-truth labels and model predictions vary based on task. For `Step-Level`, we aggregate all steps and all verifier predictions across all responses, whereas for `Response-Level` and `ErrorID`, we compute metrics at the response level. These metrics quantify verifier behavior in terms of correctly identifying mistakes versus correct answers. Balanced Accuracy and Balanced F1 both serve as aggregate measures: the former reflects average performance across both modes, while the latter penalizes imbalanced performance. An ideal verifier scores highly on both.

---

[2]This is equivalent to the "F1 Score" used by ProcessBench, which differs from the typically used F1 Score by using TNR instead of precision. To avoid confusion, we denote this metric Balanced F1 Score.

Table 2: Main evaluation results on Hard2Verify across our three evaluation tasks (§ 3.6). We report Balanced Accuracy and Balanced F1 Score. **Best** and second-best scores in each category marked.

| | Step-Level | | Response-Level | | ErrorID | |
|---|---|---|---|---|---|---|
| | Bal. Accuracy | Bal. F1 | Bal. Accuracy | Bal. F1 | Bal. Accuracy | Bal. F1 |
| *Generative Critics*, *proprietary models* | | | | | | |
| GPT-5 | **83.60** | **83.33** | 76.61 | 73.43 | 60.02 | **59.92** |
| Gemini 2.5 Pro | 82.85 | 82.74 | 74.42 | 69.83 | 48.14 | 47.45 |
| Claude Sonnet 4 | 71.97 | 62.93 | 78.74 | 76.67 | 55.49 | 40.07 |
| GPT-5-Mini | 80.93 | 79.29 | 77.44 | 76.93 | 60.55 | 58.04 |
| o3 | 77.95 | 74.99 | 76.93 | 76.91 | **61.32** | 56.81 |
| o4-Mini | 75.22 | 70.57 | **80.38** | **79.26** | 58.33 | 46.37 |
| GPT-4.1 | 59.39 | 33.60 | 62.77 | 40.68 | 53.35 | 23.55 |
| *Generative Critics*, *large (≥ 70B) models* | | | | | | |
| Kimi K2 | 64.19 | 46.46 | 68.66 | 55.85 | 51.20 | 43.39 |
| DeepSeek-R1 | 73.75 | 68.37 | 71.67 | 69.86 | 54.53 | 48.35 |
| Qwen3-235B-A22B | 71.85 | 62.57 | 74.21 | 71.60 | 56.36 | 48.08 |
| Qwen3-Next-80B-A3B | 70.09 | 59.45 | 75.77 | 73.16 | 51.33 | 42.29 |
| Qwen2.5-72B-Instruct | 52.12 | 13.69 | 57.09 | 24.84 | 45.98 | 4.16 |
| GLM-4.5-Air | 61.73 | 41.12 | 66.82 | 56.40 | 48.62 | 24.23 |
| gpt-oss-120B | **80.50** | **78.38** | **82.17** | **82.16** | **60.36** | **59.81** |
| Llama-3.3-70B-Instruct | 54.43 | 18.72 | 56.82 | 28.94 | 49.01 | 2.80 |
| *Generative Critics*, *small/medium (< 70B) models* | | | | | | |
| Qwen3-32B | 63.83 | 47.68 | 68.38 | 58.08 | 54.14 | 30.60 |
| Qwen3-30B-A3B | 66.42 | 53.24 | 70.59 | 64.37 | 55.36 | 41.50 |
| ByteDance Seed-OSS-36B | 69.19 | 59.64 | 68.67 | 62.16 | **56.50** | **47.00** |
| gpt-oss-20B | **75.43** | **71.11** | **78.27** | **78.14** | 39.84 | 39.83 |
| Qwen3-14B | 63.31 | 43.83 | 68.30 | 55.12 | 49.19 | 28.15 |
| Qwen3-8B | 60.19 | 36.44 | 62.78 | 49.63 | 46.92 | 24.08 |
| Qwen2.5-14B-Instruct | 52.44 | 27.91 | 62.38 | 58.81 | 45.92 | 13.09 |
| Qwen2.5-7B-Instruct | 51.71 | 15.42 | 52.15 | 29.60 | 35.04 | 10.43 |
| *Process Reward Models*, *open-source models* | | | | | | |
| Qwen2.5-Math-PRM-72B | 54.89 | 30.46 | **66.54** | **65.76** | **43.85** | **38.38** |
| Qwen2.5-Math-PRM-7B | 56.37 | 44.58 | 60.64 | 45.67 | 28.73 | 28.62 |
| Skywork-PRM-7B | 44.67 | 21.39 | 53.47 | 46.33 | 21.33 | 13.89 |
| Skywork-PRM-1.5B | 42.41 | 14.36 | 51.69 | 16.31 | 9.23 | 9.22 |
| ReasonFlux-PRM-7B | 52.84 | 19.79 | 54.60 | 50.54 | 42.19 | 24.53 |
| UniversalPRM-7B | **61.61** | **57.31** | 56.39 | 44.51 | 26.96 | 26.49 |

## 4.2 EVALUATED MODELS

We select a variety of PRMs and generative models prompted as step-level critics. For PRMs, we select Qwen2.5-Math-PRM-{7B,72B} (Zhang et al., 2025b), Skywork-PRM-{1.5B,7B} (He et al., 2024b), ReasonFlux-PRM-{1.5B,7B} (Zou et al., 2025), and UniversalPRM (Tan et al., 2025). We tune PRM thresholds following Zheng et al. (2024); See App. F.1 for more details. For prompted generative critics, we test a closed-source frontier models as well as large (> 70B) and small-medium (<70B) open-weight models. We evaluate all reasoning models at their maximum provided reasoning level (e.g., "high" for GPT-5), using suggested sampling parameters for various baselines. All Qwen3 models are evaluating with "thinking on". For instruction-tuned models, we use greedy decoding. The full set of models is enumerated in App. F.

## 4.3 MAIN EVALUATION RESULTS

Table 2 presents our main results, with detailed results presented in App. C. Among proprietary models GPT-5 stands out in its overall ability, able to accurately provide step-level correctness labels and identify the first error in reasoning. Gemini 2.5 Pro follows closely for step-level identification, but lags in error identification. Interestingly, o4-Mini operates the best at response level; Table 4 in App. C shows o4-Mini has relatively high and balanced TPR and TNR at the response level compared to GPT-5, which tends to be overly critical for correct responses.

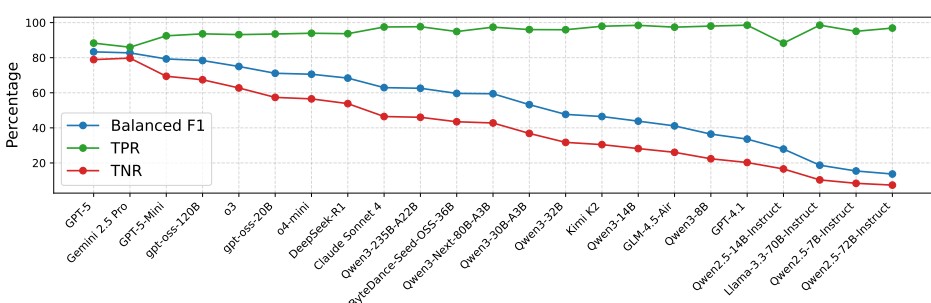

Figure 4: *Weaker models are unable to find mistakes, eventually considering* all *steps correct: TNR tends toward 0 while TPR tends towards 1.*

Among larger open-weight models, the gpt-oss series are clear standouts, with gpt-oss-120B beating GPT-5 in terms of well-rounded behavior. Recent larger Qwen3 models and DeepSeek-R1 challenge for second place. Notably, Llama-3.3-70B, which performs admirably on ProcessBench (Fig. 1) performs extremely poorly, achieving only 2.80 on Balanced F1 for the error identification task.

Among smaller models, gpt-oss-20B performs extremely well on step-level and response-level tasks, but falters in identifying errors. ByteDance Seed-OSS-36B and Qwen3-30B-A3B are the next best performers, but barely perform above random guessing levels (50%) in error identification.

Finally, even state-of-the-art PRMs, like the Qwen2.5-Math-PRM series struggle immensely on *Hard2Verify*, performing significantly below random guess performance in error identification. For example, in terms of balanced accuracy, Qwen2.5-72B achieves only 43.85%.

**What separates strong verifiers from weak verifiers?** To provide additional insights into performance variations across different models, Fig. 4 plots the TPR and TNR for all generative critics models, sorted in performance from weakest (left) to strongest (right) in terms of Balanced F1 Score. A clear trend emerges: Verifier performance degrades because TNR drops quickly to 0, while TPR rises gradually to 1. This indicates that all steps are labeled as correct, revealing that *weaker verifiers cannot catch errors*. Notably, the order of models from left to right approximately correlates with mathematical *generation* ability, i.e., the ability to solve extremely difficult math problems. As such, this may indicate that a baseline level of solving ability is a necessary prerequisite for verification. App. C.2 shows this trend holds similarly for `Response-Level` and `ErrorID` tasks.

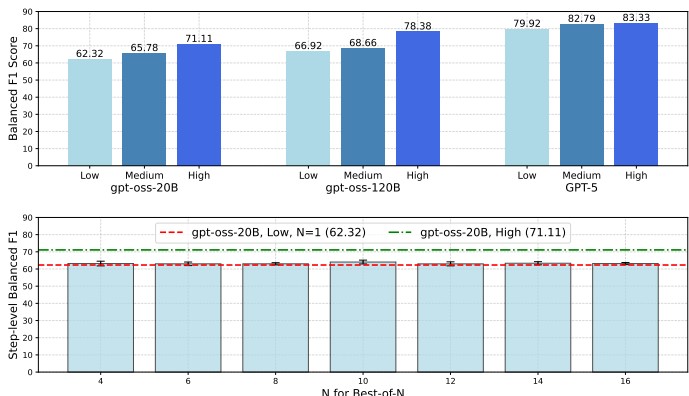

Figure 5: **Top**: Scaling inference-time compute sequentially leads to higher performance in GPT-5 and gpt-oss models, with large gains for gpt-oss-20B (62.32→71.11) and 120B (66.92 → 78.38) in terms of step-level Balanced F1. **Bottom**: Parallel decoding has little effect on step-level F1 performance for gpt-oss-20B, failing to bridge the gap vs. gpt-oss-20B at high-reasoning effort.

# 5 ADDITIONAL ANALYSIS

## 5.1 HOW SHOULD WE SCALE VERIFIER INFERENCE-TIME COMPUTE?

Here, we experiment with scaling verifier inference-time compute along via sequential and parallel approaches. We find sequential scaling brings substantive gains, whereas parallel scaling does not.

**Sequential inference-time compute scaling.** Here we explore scaling inference-time compute sequentially by letting the verifier output more tokens when verifying, focusing on the `Step-Level` task. We use gpt-oss-20B, gpt-oss-120B, and GPT-5, which all have three distinct reasoning levels: low, medium, and high. In Fig. 5 (top), we plot Balanced F1. Affording the verifier to generate more "thinking" tokens at inference time generally improves performance, with gpt-oss-120B improving the most from low (66.92) to high (78.38) and gpt-oss-20B likewise improving significantly. Gains for GPT-5 are more mild, with approximately 5% relative improvement from low to high.

**Parallel inference-time compute scaling.** Here, we attempt to match the performance of gpt-oss-20B at high reasoning effort by sampling $N$ outputs in parallel from gpt-oss-20B at low reasoning effort. To do so, we sample 32 responses per sample from gpt-oss-20B and aggregate predicted step-level labels via majority vote, breaking ties arbitrarily. Fig. 5 (bottom) shows the mean and standard deviation of 10 bootstrap sampling trials for each $N$, sweeping $N$ from 4 to 16. We also plot the baseline gpt-oss-20B performance at low and high reasoning efforts. Surprisingly, Best-of-$N$ does not meaningfully improve over sampling 1 response as $N$ increases. An intuitive explanation for this phenomenon is that step-level verification is inherently a sequential task: Each step must be processed one-after-another. As such, affording the verifier more time to "think" about each step is more effective than sampling multiple "rushed" judgments.

## 5.2 HOW DO VERIFIERS VERIFY THEIR OWN RESPONSES?

Here, we investigate the dynamics of self-verification, focusing on GPT-5, Gemini 2.5 Pro, and Claude Sonnet 4 as verifiers. Fig. 6 plots the step-level TPR and TNR performance based on response generator. The results here notably depend on verifier strength: Table 2 show that GPT-5 and Gemini 2.5 Pro are the top two performers, whereas Claude Sonnet 4 is a relatively weak proprietary verifier.

For GPT-5 and Gemini 2.5 Pro, TPR tends to correlate inversely with model strength. Both models can reliably identify correct steps from the weaker Claude Sonnet 4, but begin to label correct steps as incorrect as generator strength increases. Claude Sonnet 4, on the other hand, overwhelmingly assigns "Correct" as a label, leading to high TPRs regardless of generator. The TNR of GPT-5 and Gemini 2.5 Pro reveal that catching errors in self-produced solutions is harder than finding errors in other solutions, but the degree of difficulty depends on the model. For GPT-5, the TNR on GPT-5 generated responses is nearly 8.6 points lower that on other generators, whereas for Gemini, the drop is less than 2 points. This result is consistent with recent work analyzing self-reflection (Stechly et al., 2023; 2024; Huang et al., 2023), where LLMs were shown to have difficulties correcting their

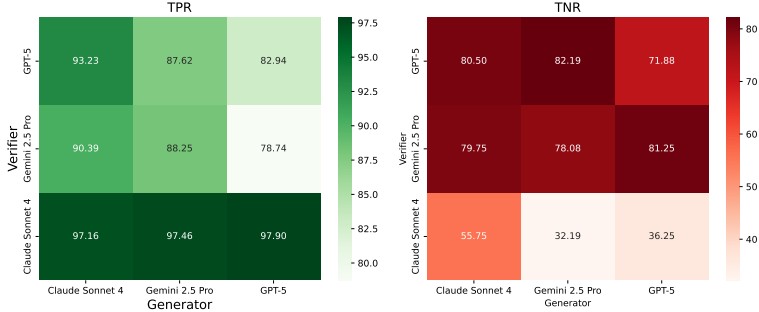

Figure 6: Verifier TPR and TNR based on response generator model. For strong verifiers (GPT-5, Gemini 2.5 Pro), TPR is inversely correlated with generator strength, while TNR is lower for self-generated responses. The latter indicates difficulties in catching self-generated errors.

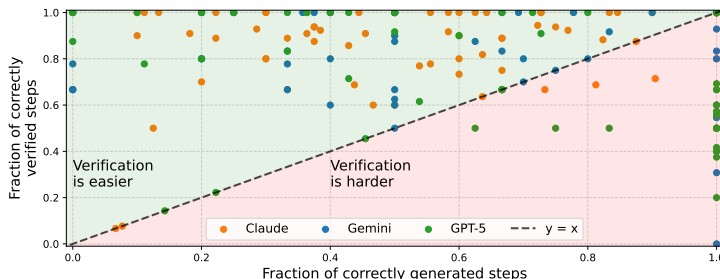

Figure 7: Each generators evaluates self-produced responses, and the fraction of steps correctly solved vs. fraction of steps correctly verified for a given question is plotted. In general, we find that models are more successful in catching mistakes than generating error-free responses. However, this becomes harder given fully correct solutions.

own mistakes in challenging reasoning settings. In contrast, Claude Sonnet 4 as a relatively weaker verifier cannot identify errors in stronger model responses.

### 5.3 IS VERIFYING PROBLEMS EASIER THAN SOLVING PROBLEMS?

Here, we examine if generating a solution is easier than verifying the same solution. We split Hard2Verify into three subsets corresponding to each of the three generator models and have the generators verify their own responses. For each response, we record the fraction of correctly generated steps ("solve rate"), as deemed by human annotators, and the fraction of correctly verified steps ("verification rate"), as deemed by agreement with human annotators. In Fig. 7, we plot the verification rate against the solve rate. We observe that the verification rate is consistently higher than the solve rate across all models when the solve rate is relatively low, i.e., $< 0.6$. However, for problems that are correctly solved, we find that verification becomes *harder*. Notably, this trend is seen typically for the stronger GPT-5 and Gemini 2.5 Pro, directly reflecting the TPR trends in § 5.2.

### 5.4 CASE STUDY: WHERE DO MODELS AND HUMANS DISAGREE?

We inspect outputs from a relatively strong open-source verifier, ByteDance Seed-OSS-36B (Team, 2025) on multiple IMO-level problems and found a recurring theme: The verifier incorrectly accepts *partial or under-justified claims as correct*. We provide two concrete examples below. These mismatches reflect larger *systematic* behavior in verifiers, revealed in § 4: Current verifiers are too *generous*, with TPR rate tending towards 1 and TNR tending toward 0, indicating that a vast majority steps are considered correct.

On IMO 2023 Shortlist, question A6, Gemini 2.5 Pro makes a generalized claim, but only proves the claim for a single input. Human annotators catch this mistake, noting "*The equality holds only at one point ... not a polynomial identity, so coefficients need not match.*" Seed-OSS-36B considers this step correct without mentioning the unfounded generalization. Similarly, on IMO 2024 Shortlist, question A1, Claude Sonnet 4 as generator constructs a proof by cases by invoking Weyl's equidistribution theorem, but considers only a single case: "*if $\alpha$ is not an even integer, then $\alpha = m + \beta$ with $m$ odd and $2/3 \leq \beta < 1$...*". Seed-OSS-36B greenlights this step as correct, whereas human annotators find it incomplete: "*The case analysis ignores the branch where $m$ is even and $0 < \beta < 1/3$,...*". Further, the theorem invocation itself is deemed under-specified: "*justification [for invoking Weyl's equidistribution theorem] should explicitly specify the estimate and the choice of $n$*".

## 6 CONCLUSION

We introduce *Hard2Verify*, a human-annotated, step-level benchmark aimed to assess how step-level verifiers operate in frontier settings. We focus our data curation on recent open-ended math problems, sampling responses from frontier LLMs. The end result of over 500 hours of human annotation effort is a benchmark that challenges many current open-source verifiers, which are unable to match the performance of larger, proprietary models.

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

APPENDIX

## A STATEMENTS

**Use of LLMs.** We minimally used LLMs during the writing process, only to check for grammar, spelling, and writing mistakes.

**Reproducibility Statement.** We plan to open-source our dataset and will ensure the release is well-documented.

## B DETAILED DATASET SOURCES

In table App. B we provide the distribution of the 80 problems we sourced from different olympiads along with the date the olympiads were conducted. For the IMO-shortlist, it is the earliest date the set of questions were made public.

| Contest | Date of Olympiad | Number of Q |
|---|---|---|
| IMO - Shortlist 2023 | 21 July 2024 | 10 |
| IMO - Shortlist 2024 | 23 July 2025 | 29 |
| Putnam | 7 Dec 2024 | 12 |
| EGMO (European Girls' Mathematical Olympiad) | 17 April 2025 | 6 |
| IMO (International Mathematical Olympiad) | 20 July 2025 | 6 |
| BMO (British Mathematical Olympiad) | 22 Jan 2025 | 4 |
| CMO (Canadian Mathematical Olympiad) | 6 March 2025 | 4 |
| USA-JMO (Junior Mathematical Olympiad) | 20 March 2025 | 4 |
| INMO (Indian National Mathematical Olympiad) | 19 Jan 2025 | 3 |
| USAMO (United States of America Mathematical Olympiad) | 20 March 2025 | 2 |
| **Total** | | **80** |

Table 3: Distribution of questions from various Olympiads with Year-wise Splits

## C ADDITIONAL EXPERIMENTAL RESULTS

### C.1 COMPLETE EXPERIMENTAL RESULTS

We report TPR and TNR for all evaluated models in Table 4, alongside our aggregate metrics presented in § 4.

### C.2 ADDITIONAL TPR AND TNR RESULTS FOR OTHER TASKS

Here, we demonstrate that the `Step-Level` trends in TPR and TNR shown in Fig. 4 hold for other tasks as well. Fig. 8 show that TNR is the primary driver in poor Balanced F1 performance: The weaker the verifier, the more it struggles in identifying mistakes, opting to mark every step as correct.

## D PROMPTS FOR GENERATION AND EVALUATION

In this section we provide the prompts used for querying frontier models to generate responses for the olympiad questions, as well as for evaluation of the solutions by verifier for step level and ErrorID task.

Table 4: Complete metrics for our three evaluation tasks, reporting Balanced Accuracy, Balanced F1, TPR, and TNR.

| | Step-Level | | | | Response-Level | | | | ErrorID | | | |
|---|---|---|---|---|---|---|---|---|---|---|---|---|
| | Bal. Accuracy | Bal. F1 | TPR | TNR | Bal. Accuracy | Bal. F1 | TPR | TNR | Bal. Accuracy | Bal. F1 | TPR | TNR |
| *Generative Critics, proprietary models* | | | | | | | | | | | | |
| GPT-5 | 83.60 | 83.33 | 88.30 | 78.90 | 76.61 | 73.43 | 61.02 | 92.20 | 60.02 | 59.92 | 57.63 | 62.41 |
| Gemini 2.5 Pro | 82.85 | 82.74 | 85.96 | 79.75 | 74.42 | 69.83 | 55.93 | 92.91 | 48.14 | 47.45 | 42.37 | 53.90 |
| Claude Sonnet 4 | 71.97 | 62.93 | 97.49 | 46.46 | 78.74 | 76.67 | 91.53 | 65.96 | 55.49 | 40.07 | 84.75 | 26.24 |
| gpt-5-mini | 80.93 | 79.29 | 92.46 | 69.41 | 77.44 | 76.93 | 71.19 | 83.69 | 60.55 | 58.04 | 72.88 | 48.23 |
| o3 | 77.95 | 74.99 | 93.15 | 62.75 | 76.93 | 76.91 | 77.97 | 75.89 | 61.32 | 56.81 | 77.97 | 44.68 |
| o4-mini | 75.22 | 70.57 | 93.93 | 56.52 | 80.38 | 79.26 | 89.83 | 70.92 | 58.33 | 46.37 | 84.75 | 31.91 |
| gpt-4.1 | 59.39 | 33.60 | 98.53 | 20.25 | 62.77 | 40.68 | 100.00 | 25.53 | 53.35 | 23.55 | 93.22 | 13.48 |
| *Generative Critics, large ($\geq$ 70B) models* | | | | | | | | | | | | |
| Kimi K2 | 64.19 | 46.46 | 97.92 | 30.45 | 68.66 | 55.85 | 98.31 | 39.01 | 51.20 | 43.39 | 71.19 | 31.21 |
| DeepSeek-R1 | 73.75 | 68.37 | 93.67 | 53.82 | 71.67 | 69.86 | 83.05 | 60.28 | 54.53 | 48.35 | 72.88 | 36.17 |
| Qwen3-235B-A22B | 71.85 | 62.57 | 97.66 | 46.03 | 74.21 | 71.60 | 88.14 | 60.28 | 56.36 | 48.08 | 77.97 | 34.75 |
| Qwen3-Next-80B-A3B | 70.09 | 59.45 | 97.40 | 42.78 | 75.77 | 73.16 | 89.83 | 61.70 | 51.33 | 42.29 | 72.88 | 29.79 |
| DeepSeek-R1 | 73.75 | 68.37 | 93.67 | 53.82 | 71.67 | 69.86 | 83.05 | 60.28 | 54.53 | 48.35 | 72.88 | 36.17 |
| Qwen2.5-72B-Instruct | 52.12 | 13.69 | 96.88 | 7.37 | 57.09 | 24.84 | 99.60 | 14.18 | 45.98 | 4.16 | 89.83 | 2.13 |
| GLM-4.5-Air | 61.73 | 41.12 | 97.40 | 26.06 | 66.82 | 56.40 | 93.22 | 40.43 | 48.62 | 24.23 | 83.05 | 14.18 |
| openai-gpt-oss-120b | 80.50 | 78.38 | 93.59 | 67.42 | 82.17 | 82.16 | 81.36 | 82.98 | 60.36 | 59.81 | 66.10 | 54.61 |
| Llama-3.3-70B-Instruct | 54.43 | 18.72 | 98.53 | 10.34 | 56.82 | 28.94 | 96.61 | 17.02 | 49.01 | 2.80 | 96.61 | 1.42 |
| *Generative Critics, small/medium (< 70B) models* | | | | | | | | | | | | |
| Qwen3-32B | 63.83 | 47.68 | 95.93 | 31.73 | 68.38 | 58.08 | 94.92 | 41.84 | 54.14 | 30.60 | 89.83 | 18.44 |
| Qwen3-30B-A3B | 66.42 | 53.24 | 96.01 | 36.83 | 70.59 | 64.37 | 91.53 | 49.65 | 55.36 | 41.50 | 83.05 | 27.66 |
| ByteDance Seed-OSS-36B | 69.19 | 59.64 | 94.89 | 43.48 | 68.67 | 62.16 | 89.83 | 47.52 | 56.50 | 47.00 | 79.66 | 33.33 |
| openai-gpt-oss-20b | 75.43 | 71.11 | 93.50 | 57.37 | 78.27 | 78.14 | 81.36 | 75.18 | 39.84 | 39.83 | 40.68 | 39.01 |
| Qwen3-14B | 63.31 | 43.83 | 98.44 | 28.19 | 68.30 | 55.12 | 98.31 | 38.30 | 49.19 | 28.15 | 81.36 | 17.02 |
| Qwen3-8B | 60.19 | 36.44 | 98.01 | 22.38 | 62.78 | 49.63 | 91.53 | 34.04 | 46.92 | 24.08 | 79.66 | 14.18 |
| Qwen2.5-14B-Instruct | 52.44 | 27.91 | 88.30 | 16.57 | 62.38 | 58.81 | 47.46 | 77.30 | 45.92 | 13.09 | 84.75 | 7.09 |
| Qwen2.5-7B-Instruct | 51.71 | 15.42 | 95.03 | 8.39 | 52.15 | 29.60 | 86.44 | 17.86 | 35.04 | 10.43 | 64.41 | 5.67 |
| *Proces Reward Models, open-source models* | | | | | | | | | | | | |
| Qwen2.5-Math-PRM-72B | 54.89 | 30.46 | 91.51 | 18.27 | 66.54 | 65.76 | 59.32 | 73.76 | 43.85 | 38.38 | 59.32 | 28.37 |
| Qwen2.5-Math-PRM-7B | 55.04 | 37.46 | 86.14 | 23.94 | 59.92 | 52.60 | 38.98 | 80.85 | 31.55 | 29.80 | 38.98 | 24.11 |
| Skywork-PRM-7B | 42.94 | 39.30 | 55.45 | 30.43 | 51.39 | 19.19 | 10.71 | 92.06 | 11.31 | 11.28 | 10.71 | 11.9 |
| Skywork-PRM-1.5B | 42.41 | 14.36 | 76.91 | 7.92 | 51.69 | 16.31 | 8.93 | 94.44 | 9.23 | 9.22 | 8.93 | 9.52 |
| ReasonFlux-PRM-7B | 52.84 | 19.79 | 94.63 | 11.05 | 54.60 | 50.54 | 69.49 | 39.72 | 42.19 | 24.53 | 69.49 | 14.89 |
| UniversalPRM-7B | 61.61 | 57.31 | 77.90 | 45.33 | 56.39 | 44.51 | 30.51 | 82.27 | 26.96 | 26.49 | 30.51 | 23.40 |

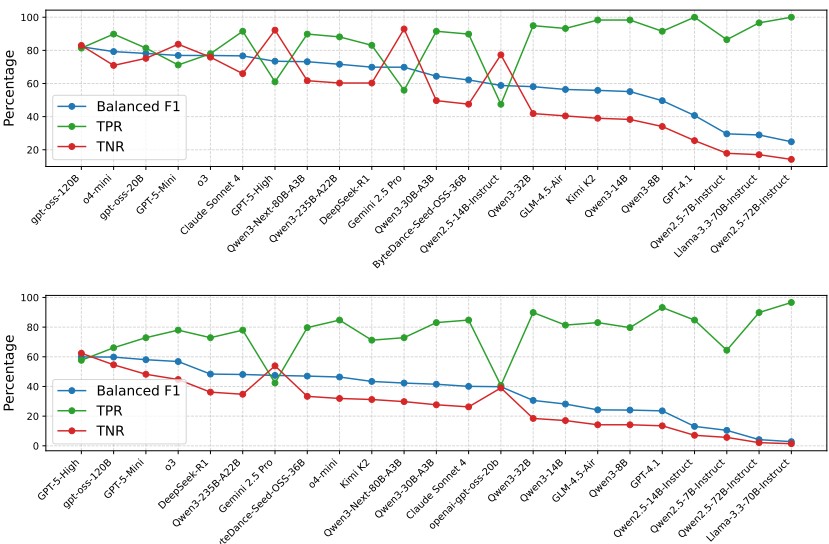

Figure 8: `Response-Level` and `ErrorID` tasks follow similar trends in TPR and TNR, with weaker verifiers unable to identify mistakes.

**Prompt used to generate responses to olympiad questions**

```
You are a careful, rigorous math proof assistant.  Provide correct,
detailed, and complete proofs.
```

Solve the following math problem formally. Return a detailed and formal solution that can be verified by a grader.

Use start the proof with `<start>` followed by each step with `<step>...</step>`, and end with `<end>`.

Only return the solution, in detailed steps, no headers, no explanations, no other text, only the `<start>` `<step>...</step>` `<step>...</step>` ... `<end>` tags.

## Prompt used for evaluating step level solutions

The following is a math problem and a solution (split into steps, enclosed with tags and indexed from 0):

[Math Problem]

problem

[Solution]

steps

Your task is to review and critique the solution step-by-step.

For each step, determine if it is correct or incorrect. A correct step is one where all of the content is correct, and is logically consistent with all previous steps and information given in the problem.

An incorrect step is one where the content is incorrect, or is not logically consistent with all previous steps and information given in the problem, or is based on an error in a previous step.

Important: Any step that contains or is based on an error is considered incorrect. That is, if the error is carried forward from a previous step or is based on an error in the previous step, consider the step incorrect.

Provide reasoning for your correctness determinations. Your final verdict should be a comma-separated list of yes and no's, where each yes or no corresponds to a step's correctness, with yes meaning correct and no meaning incorrect.

Please use the following format to return your answer:

Reasoning: <your reasoning for each step>

Verdict: <your comma-separated list of yes and no's>

Do not use any other formatting, including markdown, bold text, code blocks, or any other formatting. If your formatting is incorrect, your evaluation will be affected.

## Prompt used for Step-level ERROR ID task

The following is a math problem and a solution (split into steps, enclosed with tags and indexed from 0):

[Math Problem]

problem

[Solution]

steps

Your task is to identify the first incorrect step in the solution.

Instructions:

- Review each step carefully to determine if it is correct or incorrect

```
- A correct step is one where all content is mathematically sound and
logically consistent with previous steps

- An incorrect step contains mathematical errors, logical
inconsistencies, or is based on errors from previous steps

- Find the INDEX (0-based) of the FIRST step that is incorrect

- If ALL steps are correct, return -1

Provide your reasoning and then give your final answer as a single
number in the specified format.

Please use the following format to return your answer:

Reasoning:  <your detailed reasoning explaining which steps are
correct/incorrect and why>

<prm>NUMBER</prm>

Where NUMBER is:  - The 0-based index of the first incorrect step
(e.g., 0, 1, 2, 3, ...)

- OR -1 if all steps are correct

Examples:  - If step 0 is the first incorrect step:  <prm>0</prm>

- If step 3 is the first incorrect step:  <prm>3</prm>

- If all steps are correct:  <prm>-1</prm>

Do not use any other formatting.  The PRM value must be enclosed in
<prm></prm> tags.
```

## E    ANNOTATION DETAILS

Each sample was annotated over four rounds: An initial annotation round and three rounds of reviews to resolve disagreements. A total of 52 annotators were employed for grading, with 35 having at least a graduate degree in mathematics or related fields. On average, a model response took 90 minutes to grade and 63 minutes to review, with the longest response taking up to 4 hours. Annotators were given access to external tools, such as the internet, python, Wolfram Mathematica, and LLMs strictly as assistive aids.

We present the detailed annotation guidelines provided to the math experts for step-by-step evaluation of each model solution below.

**Annotation instructions to human annotators**

```
When annotating, refer to the reference answer(s) as possible
solution(s)/proof(s).  Each question may have multiple valid
approaches, as these are open-ended questions.  The provided
reference answer(s) is an example of a valid approach; it may not
be the only such valid approach.

Base your correctness decision off of the following criteria:

Correct:  A step is considered correct if it is:

Computationally valid:  There are no mistakes in rote mathematical
operations, such as addition or computing values of known functions
(e.g., sin(pi/2))

Logically valid:  The step follows logically from previous steps
and information present in the original question.  There are no
intermediate mistakes in the reasoning.  Any and all conclusions
in the step must be logically deducible from previous correct steps.

If a step invokes any third-party mathematical results, such as
known theorems / lemmas (e.g., fundamental theorem of calculus) or
intermediate results from previous steps, then annotators must verify
that the result is used in a valid way:
```

(1) all assumptions of the result (theorem) are met

(2) the consequence of the result (theorem) is correctly described and applied to the specific problem

Important: Do not apply \Error carried forward" grading.

If a current step is derived from a previous step that is incorrect, consider the current step incorrect, even if the logic/computation of the step is correct.

Example:

Step 1: 1 + 1 = 3 [Incorrect]

Step 2: We now must add 5 to Step 1's result, which gives us 8 [Incorrect, even though the computation in the step is correct; It is based on an incorrect Step 1]

Extra note:

\Hand-waviness": If a model produces a \hand-wavy" argument, wherein they say that a new result follows by similar logic/computation as a previously established result, then annotators must verify that the hand-wavy argument in-fact holds. This means verifying (1) The previously established result's assumptions are met by the new result scenario (2) The previously established computation/logic is applicable to the new

Example:

Step N: A valid proof of Case 1, yielding Result 1

Step N+1: Case 2 follows by a similar argument to Case 1, yielding Result 2.

[This is \hand-wavy", as the exact computation is omitted by appealing to previously computed Steps]

Incorrect: A step is considered incorrect if it is:

Based in any way on an incorrect past step.

Logically invalid: The model's output contains a reasoning error or mistake. Examples: Unfounded logical leap

Incorrectly invoking a mathematical result or past result when assumptions/conditions are not satisfied

Incorrect application of a mathematical result when conditions are met, i.e., mis-applying a theorem.

Failing to consider/cover a scenario or case within a proof, i.e., the proof concludes without covering all scenarios and is incomplete.

If the top-level proof misses a case/scenario: As this case involves text not in the model output, there is no concrete step to mark as incorrect. As a result, mark the conclusion of the proof (i.e., last step) as incorrect and provide corresponding justification.

If an intermediate result is stated, but the derivation of the intermediate result misses a case/scenario: Mark the step that states the intermediate result as incorrect (as well as any subsequent steps that depend on the intermediate result). As a concrete toy example Say a model is doing Proof by Cases for all real numbers.

It splits its analysis into 2 cases, Case 1 (positives) and Case 2 (negatives). For Case 1, it proves the claim for all positive integers, but does not consider non-integer reals.

Mark the step that contains the conclusion of Case 1 incorrect, as well as any subsequent steps that depend on Case 1.

Computationally invalid: Makes an operation / value computation mistake. This should be relatively easy to spot, but please verify

```
all complex expressions, such as integrals, trigonometric functions,
etc.
Note:  This is not an exhaustive list of errors.  Verify all
computations, and document any error that occurs, no matter how
minor.
```

# F    EVALUATED BASELINES

Here we provide a comprehensive list of models that were evaluated on our benchmark.

- OpenAI: GPT-5, GPT-5-Mini (OpenAI, 2025a), o3, o4-Mini (OpenAI, 2025b), GPT-4.1 (OpenAI, 2025), gpt-oss-120b, gpt-oss-20b (Agarwal et al., 2025)
- Google: Gemini 2.5 Pro (Google, 2025a)
- Anthropic: Claude Sonnet 4 (Anthropic, 2025)
- Moonshot (Kimi): Kimi K2 (Team et al., 2025)
- DeepSeek: DeepSeek-R1 (Guo et al., 2025)
- Alibaba Qwen:  Qwen3-235-A22B, Qwen3-Next-80B-A3B, Qwen3-32B, Qwen3-30B-A3B, Qwen3-14B, Qwen3-8B (Yang et al., 2025), Qwen2.5-72B-Instruct, Qwen2.5-14B-Instruct, Qwen2.5-7B-Instruct (Team, 2024), Qwen2.5-Math-PRM-72B, Qwen2.5-Math-PRM-7B (Zhang et al., 2025b)
- Zhipu GLM: GLM-4.5-Air (Zeng et al., 2025)
- Meta: Llama-3.3-70B-Instruct (Grattafiori et al., 2024)
- ByteDance: ByteDance Seed-OSS-36B (Team, 2025)
- Skywork: Skywork-PRM-7B, Skywork-PRM-1.5B (He et al., 2024b)
- ReasonFlux-PRM-7B (Zou et al., 2025)
- UniversalPRM-7B (Tan et al., 2025)

## F.1    PRM THRESHOLD TUNING

To decide the cutoff threshold for evaluated PRMs, we select 100 responses at random from our benchmark and tune PRM performance against this subset, following (Zheng et al., 2024). The same 100 responses are kept fixed across all baselines, and we sweep the threshold from 0.1 to 0.9 in increments of 0.05. To select the threshold, we compute the harmonic mean of the three task-specific Balanced F1 Scores, prioritizing selecting a threshold that yields strong yet balanced performance. We find that PRM performance can vary considerably based on chosen threshold.

