# OpenReview forum: "Hard2Verify: A Step-Level Verification Benchmark for Open-Ended Frontier Math"
_ICLR.cc/2026/Conference — ICLR 2026 Conference Withdrawn Submission_

### Official Review · Reviewer_FLHa · 2025-10-27

**Soundness:** 2
**Presentation:** 1
**Contribution:** 2
**Rating:** 2
**Confidence:** 5

**Summary:**

This paper introduces Hard2Verify, a human-annotated step-level verification benchmark for open-ended frontier math, created with over 500 hours of human labor. It assesses verifiers on frontier LLM responses to recent, hard math questions. The authors evaluate 29 models, finding open-source verifiers lag behind closed-source ones. They also analyze factors like poor verification performance causes, compute scaling impacts, and self-verification dynamics.

**Strengths:**

1. The paper proposes a new benchmark, Hard2Verify, which is a human-annotated, step-level verification benchmark created with over 500 hours of human labor. It provides an excellent data resource for evaluating reward models and verifiers.
2. The experiments cover a large number of open-source and closed-source models, and conduct multi-faceted analyses based on the evaluation results, offering in-depth insights into the performance of step-level verifiers in frontier mathematical reasoning scenarios.

**Weaknesses:**

1. The abstract mentions that models' answers to IMO questions require step-by-step and rigorous evaluation, and training such verifiers is highly challenging—a widely recognized issue. However, the connection between this point and the proposed Hard2Verify benchmark in the paper is not clearly elaborated, making the transition and coherence abrupt and awkward, which is quite misleading.

2. In my understanding, the vast majority of RLVR works conduct RL training through outcome reward, which is also one of the reasons for DeepSeek-R1's success. At the stage where process reward has not been widely used in RLVR, the paper fails to provide sufficient evidence to assert that "the next frontier for LLMs is solving problems that are hard to verify".

3. Following the previous issue, although the experiments cover many models, the reward models are still mainly PRMs, which are not suitable for current RL training. It is believed that results of more RLVR verifiers (such as general verifier, xverify, R1-Distill-Verifier) should be supplemented to make the evaluation more comprehensive.

4. The authors seem to have an unclear boundary between the understanding of reward models and verifiers. The paper sometimes conflates the two concepts in discussions (e.g., when analyzing model performance and application scenarios), leading to ambiguity in the logical context of related content.

**Questions:**

Please refer to the weaknesses.

---

### Official Review · Reviewer_VpB3 · 2025-11-01

**Soundness:** 3
**Presentation:** 3
**Contribution:** 3
**Rating:** 6
**Confidence:** 3

**Summary:**

The paper introduces Hard2Verify, a human-annotated, step-level verification benchmark produced with over 500 hours of human labor. Hard2Verify is designed to rigorously assess step-level verifiers at the frontier: Verifiers must provide step-level annotations or identify the first error in responses generated by frontier LLMs for very recent, challenging, and open-ended math questions.

**Strengths:**

I find the motivation of this paper quite reasonable. Currently, LLMs in solving in-formal IMO-level mathematical problems remains an open-ended challenge. The authors also propose a practical and solid benchmark for this purpose.

The paper is well-presented and easy to follow.

**Weaknesses:**

I believe the authors' proposal to use a large number of open-ended problems is a reasonable setting. However, a key issue that arises is how to ensure the quality of human annotations. I think the authors should provide more detailed information to enhance credibility. For example, did they use cross-validation among experts? What was the cross-validation accuracy? What are the backgrounds of the experts, and how many annotations were collected?

I also believe there is still a gap between verifier accuracy and actual solver improvement. The authors should include additional experiments to ensure that scores on the current benchmark truly reflect real improvements in solver performance. For instance, they could use some of the methods mentioned in the Introduction section to design a simple baseline, demonstrating that high-scoring verifiers in Table 2 can indeed help improve solver scores.

**Questions:**

Why use responses from only three models as the benchmark data? How can we ensure that the distribution of model-generated responses is sufficiently diverse and reasonable?

---

### Official Review · Reviewer_m8mN · 2025-11-02

**Soundness:** 3
**Presentation:** 3
**Contribution:** 1
**Rating:** 2
**Confidence:** 4

**Summary:**

This paper proposes Hard2Verify, a step-level verification benchmark centered on IMO mathematics reasoning questions. By generating reasoning responses using frontier close-source language models (gpt/claude/gemini) and annotations manually with the help of human math experts (>500 hours of human labor), the benchmark focuses on metrics including step-level and response-level accuracy,  and errorID. Finally, this work evaluates a wide range of general LLMs and PRMs on the benchmark and presents insights on verification-compute scaling, self-verification biases and difficulty comparison between verficiation and solving using LLMs.

**Strengths:**

1.The paper is well written, clear, and most claims are well supported by the experimental evidence.

2.The evaluation is comprehensive, and the analysis in section 5 is insightful.

**Weaknesses:**

Despite the strengths mentioned above, the main weakness of this work, in my opinion, is the limited significance of this contribution, compared with existing work such as processbench (mentioned in the main text of the paper).
1. hard2verify includes only 80 IMO questions and 200 responses, compared with 3,400 responses in processbench, which is one order of magnitude larger;
2. one major claim in this paper is the IMO problem difficulty, however, if i understand correctly, processbench also includes omnimath and olympiadbench, both of which cover competition-level math reasoning questions. Further, processbench extends the diversity by including easier questions from gsm8k and math.
Overall, the proposed benchmark (hard2verify) is quite limited on the scale and diversity, and therefore needs significant amount of improvement.

**Questions:**

1. how do authors consider the inter-dependence of the steps? as described in the paper, all steps after a problematic step will be viewed as wrong, but how will this information be passed to the later steps when each individual step is  evaluated?
2. can the authors clarify line 244: "Note that this setting is less strict than the Step-Level setup: Exact step labels need not match exactly for a verifier to agree with a human at the response level."? what does this mean?
3. Figure 7 is very interesting. can the authors share more insights on why there are cases that generated steps are 100% correct but verification accuracy is very low, and sometimes is even close to 0? this is a bit counter-intuitive to me.

some minor points:
* line 177: "contain a mistakes" should be "contain mistakes"
* appendix E: line 922,933,943 seems to be a typo of quotation marks: e.g. \Hand-waviness" should be "Hand-waviness"
* appendix E: line 948, the list of examples is a bit hard to follow.

---

### Official Review · Reviewer_Cz4k · 2025-11-03

**Soundness:** 3
**Presentation:** 4
**Contribution:** 3
**Rating:** 6
**Confidence:** 4

**Summary:**

This paper proposes Hard2Verify, a new benchmark for step-wise verifiers in the context of advanced level math. It curates mathematical problems from international competitions (IMO, Putnam), which comprises challenging, open-ended problems that represent the frontier capabilities of recent reasoning models (GPT-5, Gemini 2.5 Pro, Claude Sonnet 4). The paper designs a careful, human expert-led step-level annotation process, comprising several review rounds on top of the responses of these reasoning models. The paper evaluates 29 different verifiers to show that Hard2Verify represents a major challenge for current models, and also provides insights on verification compute scaling, generation-verification gap, and when a qualitative analysis on the failure modes of the current models.

**Strengths:**

- The paper is well-written and easy to follow.

- The paper is very clear and transparent in the design principles that guides the benchmark creation, and the design choices are well justified.

- The benchmark is supported by a careful and transparent methodology, and the paper describes dataset curation, response generation, and annotation process in good detail.

- The benchmark comprises different evaluation modes (comprising different ways employed by the literature), as well a well-justified metric design (which is very important since step-wise annotation may potentially lead to unbalanced datasets)

- Empirically, the paper evaluates a rich set of verifiers, from different sizes and categories, which allows a good perspective on the verifiers landscape.

- Lastly, the empirical analysis and raised research questions in Sections 4 and 5 are relevant, well supported by the provided evidence and the insights are interesting for Verifiers’ research.

**Weaknesses:**

- Major: Based on the paper content, the benchmark is still not open-sourced, and therefore its content/documentation was not available for review.

- Minor: In the evaluation of verifiers (Table 2), it would be great to provide error bars to assess statistical significance of the results.

- Minor: It is somewhat unclear for how long this benchmark would be relevant, as it is based on the current state of frontier models on math tasks, which we know has evolved very rapidly in the past year or so. This does not diminish its current relevance, just open questions about its long-term impact.


Overall, the paper represents strong efforts to design a well-justified benchmark for step-level verification with good empirical insights. I believe this represents a good contribution for the current landscape of research on model-based verification. My perspective is that, from the content of the paper, it is a clear accept (score 8). Nonetheless, as the paper's major value comes from open-sourcing a benchmark for research use, I believe it is necessary to provide it during the submission/rebuttal as evidence of the work. Therefore, I am temporarily providing a score of 6 and will increase it to 8 if the authors provide a link for the data/documentation during rebuttal.

**Questions:**

N/A

---

### Note · Authors · 2026-01-05

**Comment:**

We thank the reviewers for their time and effort in reviewing our submission.

**Withdrawal Confirmation:**

I have read and agree with the venue's withdrawal policy on behalf of myself and my co-authors.